# Prediction Model for Liquid-Assisted Femtosecond Laser Micro Milling of Quartz without Taper

**DOI:** 10.3390/mi13091398

**Published:** 2022-08-26

**Authors:** Hongbing Yuan, Zhihao Chen, Peichao Wu, Yimin Deng, Xiaowen Cao, Wenwu Zhang

**Affiliations:** 1Faculty of Mechanical Engineering & Mechanics, Ningbo University, Ningbo 315211, China; 2Ningbo Institute of Materials Technology & Engineering, Chinese Academy of Sciences, Ningbo 315201, China; 3University of Chinese Academy of Sciences, Beijing 100049, China

**Keywords:** femtosecond laser, liquid assisted, design of experiment (DOE), prediction model

## Abstract

The strong nonlinear absorption effect and “cold” processing characteristics of femtosecond lasers make them uniquely advantageous and promising for the micro- and nanoprocessing of hard and brittle materials, such as quartz. Traditional methods for studying the effects of femtosecond laser parameters on the quality of the processed structure mainly use univariate analysis methods, which require large mounts of experiments to predict and achieve the desired experimental results. The method of design of experiments (DOE) provides a way to predict desirable experimental results through smaller experimental scales, shorter experimental periods and lower experimental costs. In this study, a DOE program was designed to investigate the effects of a serious of parameters (laser repetition frequency, pulse energy, scan speed, scan distance, scan mode, scan times and laser focus position) on the depth and roughness (*Ra*) of the fabricated structure through the liquid-assisted femtosecond laser processing of quartz. A prediction model between the response variables and the main parameters was defined and validated. Finally, several blind holes with a size of 50 × $50\;{\mu m}^{2}$ and a depth of 200 μm were fabricated by the prediction model, which demonstrated the good consistency of the prediction model.

## 1. Introduction

Quartz is a hard, glossy mineral consisting of slicion dioxide in crystal form. Due to its high temperature resistance, corrosion resistance, high light transmittance, thermal stability and biocompatibility, the high quality of microholes in quartz has great significance in the field of medical devices, automotive fuel injectors, and packaging. At present, there are still some technical challenges for processing microholes with small sizes, large aspect ratios, high surface quality and high dimensional accuracy. In traditional mechanical processing, the drill bit is broken easily while drilling deep holes, which leads to the low surface quality of the hole wall [1]. Many technologies have been proposed to drisll deep hole, such as electrical discharge machining (EDM), electrochemical machining (ECM), ultrasonic machining, and water jet machining. However, there are some limitations to these technologies. EDM is only suitable for conductive materials, and the processing defects are inevitable, including heat-affected zones and micro-cracks [2,3]. ECM can process conductive materials with high surface quality; the system of ECM is complex and expensive [4]. Ultrasonic machining have relative low efficiency and precision [5,6]. Water jet machining is not applicable to microfabrication due to the high pressure on the material during the fabrication [7].

Because of their ultrashort pulse widths and ultrahigh peak power, femtosecond lasers are suitable for processing almost any material, especially hard and brittle materials [8,9,10,11]. Femtosecond laser processing has many advantages, such as a small heat-affected zone, high precision and high efficiency [12,13]. However, during the femtosecond laser processing, the debris adhere in or around the processed region, which reduces the surface quality. Liquid-assisted femtosecond laser processing [14,15,16,17] provides an effective method for processing high-quality structures.

Li et al. [18] processed microholes from the rear surface of quartz by liquid-assisted femtosecond laser fabrication and found that the liquid greatly reduced the clogging of the ablation material, increasing the aspect ratio of the hole. The quality of microholes is affected by a series of parameters, such as laser power, pulse numbers, defocus amount, and pulse width. In order to improve the quality of the fabricated hole, it is necessary to study and optimize the parameters mentioned above. Usually, single-factor analysis method is used for this purpose, but it is incomprehensive. The statistical method based on design of experiments (DOE) and analysis of variance (ANOVA) analysis [19] has the ability to take into account the influence of parameters on the fabricated structure simultaneously, obtain ideal experimental results and reach correct conclusions at a lower experimental cost. Annalisa et al. [20] designed DOE experiments to estimate the effects of laser repetition rate, pulse energy, scan speed and scan distance on the depth of the removed material of polymethylmethacrylate during femtosecond laser microgrinding. The results of the study showed that the laser pulse energy is the main parameter affecting the milling depth. A predictive model describing the relationship between the depth and the main parameters was defined and validated successfully. In this study, we designed a complex DOE to investigate the effect of a series of parameters on processing quartz.

Based on the DOE and ANOVA analysis, a statistical method was provided to conduct the research on liquid-assisted femtosecond laser machining quartz. We obtained a predicted model regarding the effect of depth and roughness (*Ra*) of the fabricated structure on various parameters, including laser scan speed, scan distance, processing times, defocus amount, laser repetition, frequency, pulse energy and scan path. The experimental results demonstrated that high-quality processing of quartz can be obtained by optimizing the fabrication parameters. We processed a series of blind holes with large depth by liquid-assisted femtosecond laser drilling.

## 2. Materials and Methods

### 2.1. Materials

We chose double-sided polished quartz of a size of 10 mm × 10 mm × 0.5 mm (surface roughness 0.06 μm). The surface of the quartz was treated with alcohol by an ultrasonic cleaner to remove the particle contaminants.

### 2.2. Methods

The schematic experimental device is shown in Figure 1a. A fiber laser with pulse duration of 400 fs and central wavelength of 515 nm was selected for the experiment, as shown in Figure 1b; the laser passed the interior and focused on another surface of the quartz, which was in contact the deionized water. Figure 1c illustrated, in detail, the focus position of the laser under different defocus amounts, including negative geometric focus, 0, and positive geometric focus.

The laser-milled area was 50 μm × 50 μm. After femtosecond laser fabrication, the sample was treated by ultrasonication and cleaned with KOH solution for 5 min to remove the residue. A laser scanning confocal microscope (LSCM, Vk-x200k, Keyence, Osaka, Japan) was used to measure the depth and surface roughness of milled samples.

### 2.3. Experimentation Procedure

The statistical method based on analysis of variance (ANOVA) was selected to analyze the factors that affected the depth and surface roughness of laser milling. Moreover, the main effect diagram and two-factor interaction were used to support the technical interpretation.

Pre-design is the basis for establishing statistical methods. In the preliminary experiment, according to DOE, the range of parameters for laser milling process was studied. The laser pulse frequency was 25–300 KHz, the pulse energy was 0.1–0.8 μJ, the scan speed was 20–80 mm/s, the scan distance was 0.2–2 μm, the defocus amount was −6–6 μm and the processing time was 10–100 times. Figure 2 shows two different processing methods. The thin solid line is the laser scanning direction, the thick solid line is the contour acquisition direction, and the laser scanning back and forth along the contour acquisition direction is recorded as one cycle. Here, *d* is scan distance.

In this stage, the screen of the experiment was carried out, and the level of the process milling test was determined. We utilized a two-level full factorial design consisting of 7 parameters and 130 groups in total. The effects of each parameter involved in laser milling on the depth and *Ra* were studied. Table 1 lists the process parameters for the laser milling, maximum, minimum, and intermediate values. All groups of experiments were conducted based on the designed experiment as listed in the Table 1. We selected four groups, and the detailed femtosecond laser processing parameter values were as shown in Table 2. The three-dimensional morphology of the laser milling was as listed in Table 2 and as shown in Figure 3a–d, where the depth and *Ra* of the corresponding structure were measured by LSCM. Their depths and roughnesses were 4.69 μm, 9.88 μm, 14.83 μm, 12.3 μm and 0.21 μm, 0.27 μm, 0.27 μm, and 0.16 μm, respectively.

## 3. Results

### ANOVA Results

ANOVA was performed to assess the statistical significance of the processing factors on the response variable depth, *Ra*. The ANOVA table contains the degrees of freedom (DF), the adjusted sum of squares (Adj SS), the adjusted mean squares (Adj MS), the *F* value (*F*-value) and the *p*-value (*p*-value) for each parameter or parameter combination. During the analysis, the *p*-value determines the significance of each factor and its combination. The analysis was performed at 95% confidence level (α = 0.05). Therefore, it is considered that when the *p*-value is less than 0.05, each factor and combination are considered to be significant.

The results of the ANOVA are shown in Table 3 and Table 4. The second-order interaction factors were analyzed. The *p*-values of significant variables are highlighted in bold in the table; in order to simplify the model, the insignificant variables were deleted. The ANOVA results in Table 3 show that surface roughness is affected by the factors of pulse energy, scan speed, scan distance, scan times, and the interaction factors repetition rate × pulse energy and scan times × scan path. In Table 4, depth is affected by the interaction factors of repetition rate×pulse energy, repetition rate × scan speed, repetition rate × scan time, pulse energy × scan time, scan time × focus, and all single factors except repetition rate.

The ANOVA table cannot provide any information about the “effect” of the process parameters, but the main effect and two-factor interaction effect can be used to analyze these aspects. Figure 4 and Figure 5 show the main effect diagram and two-factor interaction effect diagram of *Ra* and depth, respectively. The main effect of factor is defined as the response change caused by the change in the factor level, which is the main factor in the research. When the corresponding difference between the levels of one factor are different at all levels of other factors, it indicates that there is interaction between factors. It can be seen from Figure 4 that the interaction effect of repetition rate × pulse energy and scan times × scan path has significant influence on the response variable *Ra* (the slope difference between the two lines is very large and very non-parallel). It can be seen from Figure 5 that the interaction effect of repetition rate × pulse energy, repetition rate×scan speed, repetition rate × scan times, pulse energy × scan times, and scan times × focus has significant influence on the corresponding variable depth (the slope difference between the two lines is large). Based on these experimental results, the experimental model is reduced according to the *p*-values in the ANOVA table. The low *p*-values indicate that the examined sample provides sufficient evidence to reject the original hypothesis for the whole aggregate. The main factors and important interactions will be part of the final predictive regression model. In addition, the curvature values in the ANOVA table were 0.245 (Table 3) and 0.786 (Table 4), indicating that the quadratic effects could be neglected and that the linear model described the behavior of the response variables well.

Regression equations defined in uncoded units were used to predict *Ra* versus depth as a function of process parameters.
(1)$$\begin{array}{c} Ra=0.4164-0.001075A-0.1566B-0.001879C+0.0302D\\ +0.000887E+0.028F+0.001723AB-0.00102EF \end{array}$$
(2)$$\begin{array}{c} Depth=10.72-0.02844A+0.66B-0.1083C-0.1772E+633F+0.0496AB\\ +0.000284AC+0.000395AE+0.1238BE+14.89EF \end{array}$$

Here *A* is the repetition rate, *B* is the pulse energy, *C* is the scan speed, *D* is the scan distance, *E* is the scan times, and *F* is the focus. The parameters expressed in equation *A*, *B*, *C*, *D*, *E*, *F* are shown in Table 1. The predictive ability of Equations (Equation 1) and (Equation 2) will be discussed in the next section.

## 4. Discussion

### 4.1. Prediction Model Validation

The regression model determined above was used to predict the confidence interval (CI) and prediction interval (PI) of the depth and *Ra* at three points in the full factor experiment. The CI and PI predicted by the software are in Table 5, and the experimental measurement results of each prediction point are also listed in Table 5 and compared with the predicted values. The experimental results show that the depth and *Ra* of all points are within the PI, and the *Ra* of all points even falls within the more stringent CI than the PI. It can be concluded that the results of the validation experiment prove the good prediction ability of the regression model for the depth and *Ra* of the fabricated structure.

### 4.2. Deep Micro Hole Machining

Based on the prediction model of DOE above, we selected a group of experimental parameters with low machining roughness (*Ra* = 0.19 μm) as follows: repetition rate 300 kHz, pulse energy 0.4 μJ, scan speed 50 mm/s, scan distance 0.5 μm, number of scan times 10, and defocus amount −0.003 mm for micro hole machining. The experiment carried out deep blind hole machining by controlling the scan times in the z-direction. The number of scan times was 67, 2 μm each time, and the depth of the processed rectangular micro hole array was 200 μm. The micro holes were split by sandpaper grinding technology, and the side wall morphology and depth of the micro holes were observed by LSCM. The experimental results are shown in Figure 6. Figure 6a demonstrates the side view of these micro holes, and Figure 6b illustrates the magnified 3D view of one micro hole. The depth of the micro hole was 200 μm. The morphology of the hole wall is shown in Figure 6c, and the roughness was 0.56 μm.

Figure 6d illustrates the cross-section of the side wall at the bottom of the blind hole, and Figure 6e depicts the cross-section of the side wall at the entrance of the micro hole. The taper angle of the micro hole was calculated by the following formula:(3)$$\begin{array}{c} \alpha =arctan((D-d)/H) \end{array}$$
where α, *D*, *d* and *H* were the taper angle, entrance width, bottom width and depth of the blind hole, respectively. In this work, the entrance width, bottom width and depth were 51.97 μm, 51.2 μm and 200 μm, respectively. The taper angle was 0.22°, which was calculated by Equation (Equation 3).

## 5. Conclusions

In this study, the liquid-assisted femtosecond laser transmissive machining quartz was investigated by the DOE. The effect of laser machining parameters (pulse energy, repetition frequency, defocus amount, scan distance, scan speed, scan times and scan path) on the response variable roughness and machining depth was assessed using ANOVA analysis. The main experimental findings are as follows:(1)DOE and ANOVA analysis can effectively screen the main factors affecting responses, avoiding large numbers of experimental studies.(2)The *Ra* is mainly influenced by the pulse energy, scan speed, scan distance, and scan times.(3)The depth is mainly affected by the pulse energy, repetition frequency, defocus amount, scan speed and scan times.(4)The effect of scan path on depth response is not obvious, but its interaction with the scan times has a more obvious effect on roughness.(5)The regression prediction model of roughness versus depth was experimentally derived and validated in a 3-point factorial scheme, with all depth values of the tests falling within the PI and all roughness values falling within a tighter CI than the prediction interval, demonstrating the good predictive power of the prediction model.(6)As high efficiency and low roughness are the first requirements of production, we used the optimized experimental parameters with a repetition frequency of 300 kHz, pulse energy 0.4 μJ, scan speed 50 mm/s, scan distance 0.5 μm, number of scan times 10, and defocus amount −0.003 mm to process large depth microblind holes.

## Figures and Tables

**Figure 1 micromachines-13-01398-f001:**
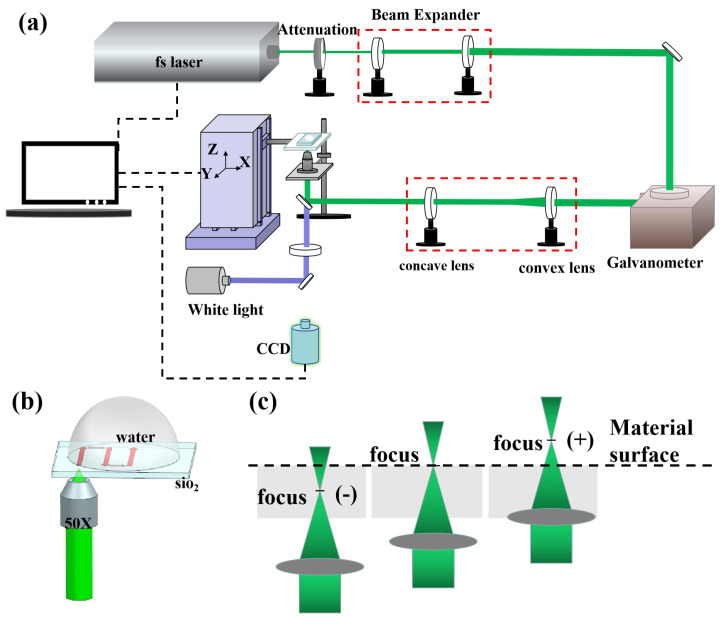
(**a**) Schematic diagram of the femtosecond laser processing system. (**b**) Enlarged view of the laser processing section. (**c**) Schematic diagram of the definition of the out-of-focus state.

**Figure 2 micromachines-13-01398-f002:**
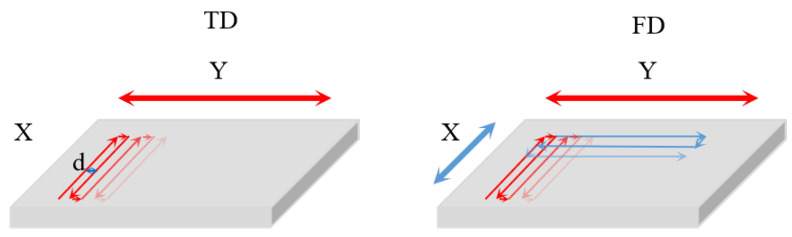
Schematic diagrams of two laser scanning patterns employed during milling of quartz samples; *d* is the distance between two consecutive scanning lines.

**Figure 3 micromachines-13-01398-f003:**
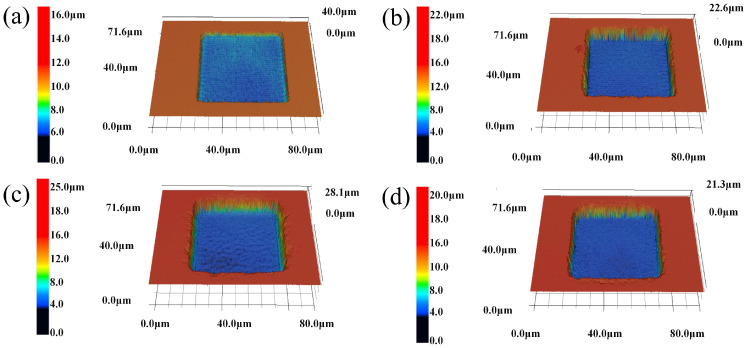
Three-dimensional morphology of some sample structures. The laser processing parameters of (**a**–**d**) are shown in Table 2.

**Figure 4 micromachines-13-01398-f004:**
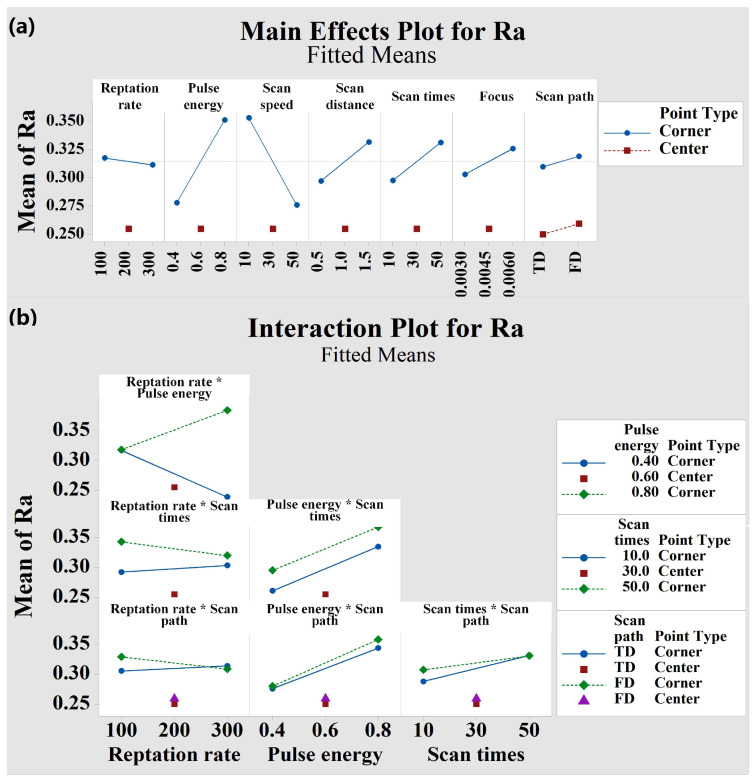
(**a**) Main effect plot of factors affecting the response variable *Ra*. (**b**) Interaction plot of factors affecting the response variable *Ra*.

**Figure 5 micromachines-13-01398-f005:**
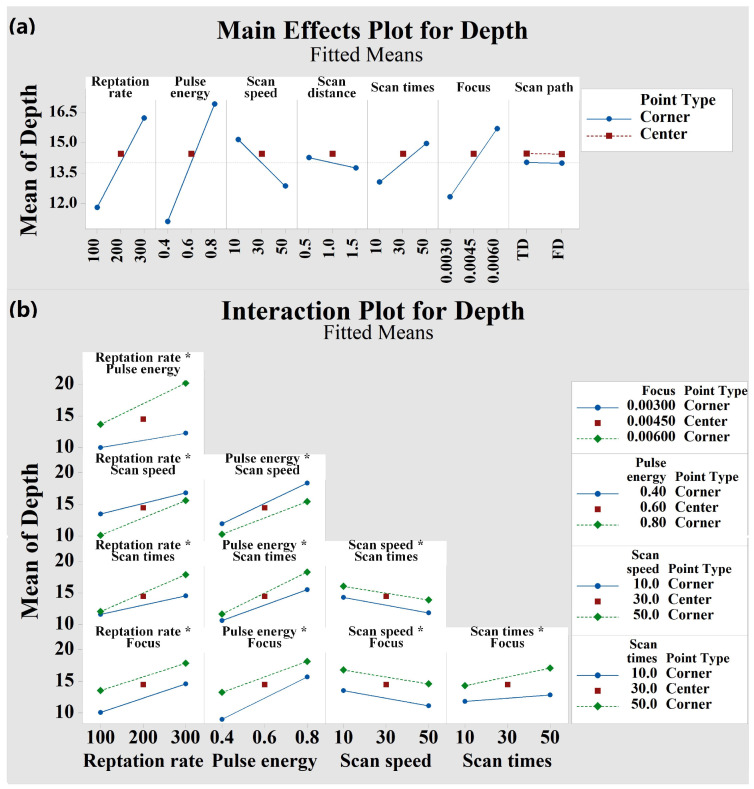
(**a**) Main effect plot of factors affecting the response variable depth. (**b**) Interaction plot of factors affecting the response variable depth.

**Figure 6 micromachines-13-01398-f006:**
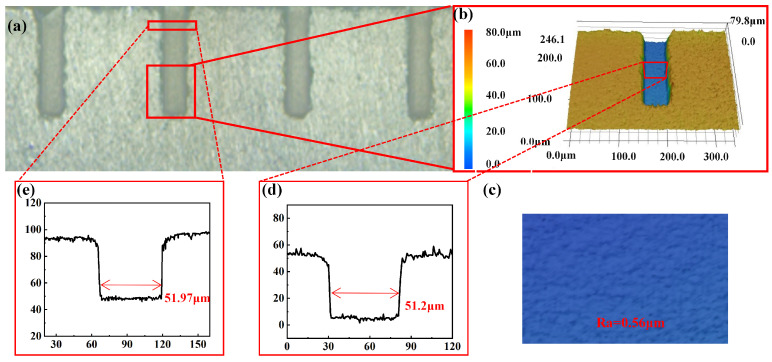
(**a**) Top view of the fabricated blind hole. (**b**) Three-dimensional local enlarged view of side section. (**c**) The enlarged view of the bottom of the blind hole. (**d**,**e**) The cross–section of the side wall.

**Table 1 micromachines-13-01398-t001:** Control factors and levels adopted in milling tests.

Factors	Abbreviation	Fs-Laser Parameters Range		
		Low Level	Central Point	High Level
Repetition rate [kHz]	*A*	100	200	300
Pulse energy [μJ]	*B*	0.4	0.6	0.8
Scan speed [mm/s]	*C*	10	30	50
Scan distance [μm]	*D*	0.5	1	1.5
Scan times	*E*	10	30	50
Focus [mm]	*F*	−0.003	−0.0045	−0.006
Scan path	*G*	TD	-	FD

**Table 2 micromachines-13-01398-t002:** Laser processing parameters of the structures in Figure 3.

	Repetition Rate [kHz]	Pulse Energy [μJ]	Scan Speed [mm/s]	Scan Distance [μm]	Scan Times	Focus [mm]	Scan Path
a	100	0.4	50	1.5	50	−0.003	FD
b	100	0.8	50	1.5	10	−0.006	TD
c	200	0.6	30	1	30	−0.0045	FD
d	300	0.4	50	1.5	50	−0.003	FD

**Table 3 micromachines-13-01398-t003:** ANOVA table for pocket *Ra* ( μm).

Source	DF	Adj SS	Adj MS	*F*-Value	*p*-Value
Model	8	0.63905	0.079881	12.84	0.000
Linear	6	0.43389	0.072315	11.63	0.000
Repetition rate	1	0.00219	0.002195	0.35	0.554
Pulse energy	1	0.18075	0.180751	29.06	**0.000**
Scan speed	1	0.18075	0.180751	29.06	**0.000**
Scan distance	1	0.02910	0.029101	4.68	**0.033**
Scan times	1	0.04026	0.040257	6.47	**0.012**
Scan path	1	0.00084	0.000838	0.13	0.714
Two-way Interactions	2	0.20516	0.102579	16.49	0.000
Repetition rate × Pulse energy	1	0.15194	0.151938	24.43	**0.000**
Scan times × Scan path	1	0.05322	0.053220	8.56	**0.004**
Error	121	0.75257	0.006220		
Curvature	1	0.00846	0.008461	1.36	0.245
Lack-of-Fit	120	0.74411	0.006201	-	-
Total	129	1.39162			

**Table 4 micromachines-13-01398-t004:** ANOVA table for pocket Depth ( μm).

Source	DF	Adj SS	Adj MS	*F*-Value	*p*-Value
**Model**	10	2514.82	251.48	38.64	0.000
Linear	5	2211.04	442.21	67.94	0.000
Repetition rate	1	600.40	600.40	92.24	**0.000**
Pulse energy	1	1044.82	1044.82	160.52	**0.000**
Scan speed	1	135.42	135.42	20.81	**0.000**
Scan times	1	94.67	94.67	14.54	**0.000**
Focus	1	335.73	335.73	51.58	**0.000**
Two-way Interactions	5	303.78	60.76	9.33	0.000
Repetition rate × Pulse energy	1	125.77	125.77	19.32	**0.000**
Repetition rate × Scan speed	1	41.36	41.36	6.35	**0.013**
Repetition rate × Scan times	1	79.73	79.73	12.25	**0.001**
Pulse energy × Scan times	1	31.38	31.38	4.82	**0.030**
Scan times × Focus	1	25.54	25.54	3.92	**0.050**
Error	119	774.59	6.51		
Curvature	1	0.48	0.48	0.07	0.786
Lack-of-Fit	118	774.10	6.56	-	-
Total	129	3289.41			

**Table 5 micromachines-13-01398-t005:** Validation of prediction model based on comparison between experimental trial results, confidence, and prediction interval identified with Minitab.

Variable	Settings of Point 1		Settings of Point 2		Settings of Point 3	
Repetition rate [kHz]	200		100		300	
Pulse energy [μJ]	0.4		0.6		0.5	
Scan speed [mm/s]	20		30		40	
Scan distance [μm]	0.5		0.8		1	
Scan times	30		45		40	
Focus [mm]	−0.003		−0.006		−0.003	
Scan path	2		4		2	
Minitab Prediction	CI	PI	CI	PI	CI	PI
Depth values [μm]	(9.206,10.812)	(4.894,15.124)	(12.848,14.778)	(8.67,18.956)	(11.863,13.819)	(7.695,17.986)
*Ra* values [μm]	(0.263,0.327)	(0.133,0.458)	(0.296,0.35)	(0.162,0.485)	(0.244,0.302)	(0.111,0.435)
Experimentation						
Trial depth value [μm]	13.05		18.21		9.65	
Trial *Ra* value [μm]	0.27		0.32		0.28	

## Data Availability

Data are contained within the article.

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
