# Peer review of "Prediction Model for Liquid-Assisted Femtosecond Laser Micro Milling of Quartz without Taper"

_micromachines, 2022, doi:10.3390/mi13091398_

Round 1

Reviewer 1 Report

This is a study on the development of a prediction model for the improvement of quartz milling that utilises design of experiments method.

The manuscript claims that the aim is to achieve laser-milling of quartz without taper, however although other topographical characteristics of the structure are investigated I have not seen evidence of investigation of the taper formation on the side walls of the machined hole.

One important parameter that affects the formation of a taper is the aspect ratio of the hole. It is normally the case that tapering is negligible in small aspect ratios, but it is becoming an issue when deep and narrow holes are required (high aspect ratios).

The question remains, what is the validity limitations on this study, and how does it apply to higher aspect ratios.

Answering this question will be a significant addition to solving this important technological problem.

For clarity, it is necessary to clearly describe the experimental parameters, i.e. scan distance, scan times, focus. On the focus in particular, are the variations referring to deviations from perfect focus? It is not clear.

Figures 2 and 3 are low quality and have small print labels that cannot be read in the PDF copy. High quality images with larger labels should be provided.

Reviewer 2 Report

The submitted paper applies design of experiments and analysis of variance to the optimization of femtosecond laser irradiation conditions for pattern depth and new bottom surface in the microfabrication of quartz crystals. However, it is difficult to recognize the significance of the following critical issues.

(1) As Table 5 shows, the CI is larger than the PI, which is incomprehensible. In addition, the CIs are too wide in general, and the data are not reliable. The contents of this paper must be regarded as having fatal problems of reliability.

(2) The results obtained from DOE and ANOVA should be used to further explore the machining parameters that are optimal for depth and roughness. There is no optimization.

(3) There is no scientific discussion on the effect of each machining parameter on the machining results, and no scientific significance is perceived. It is simply an exercise in DOE and ANOVA.

Reviewer 3 Report

The proposed manuscript within the scope of the Micromachines journal. Some of the results observations are interesting, however, if the paper can be improved in the following areas, it would add more value to the readers:

1.     Please illustrate how to select the four parameters in Table 2 for the following discussion.

2.     Please detailed illustrate results in Figures 4-5, and please magnify the figure because the legend was hard to read.

3.     There was no description for Eqns. (1)-(2). Please illustrate it.

4.     In Table 5, there was a variation between the prediction and experimental value, how to reduce the variation.

5.     Please illustrate why the only three points of each parameter were choose in regression analysis in Table I, in the experimental model, there are at least four point in each parameters to investigate the experimental trend.

Round 2

Reviewer 1 Report

I am happy with the revisions, my suggestion is that the manuscript can  ow be accepted for publication.

Reviewer 2 Report

The answers provided by the authors have helped me to generally understand the main points of this paper. I confirmed that the paper can be considered useful in the search for optimal processing parameters.